# The Effect of Perceived Stress, Family Companionship, and Mental Health on the Subjective Happiness of Chinese Healthcare Workers: A Mixed Research Method

**DOI:** 10.3390/ijerph191912058

**Published:** 2022-09-23

**Authors:** Danni Feng, Quan Wang, Sufang Huang, Xiaorong Lang, Fengfei Ding, Wei Wang

**Affiliations:** 1Tongji Hospital, Tongji Medical College, Huazhong University of Science and Technology, Wuhan 430030, China; 2School of Nursing, Tongji Medical College, Huazhong University of Science and Technology, Wuhan 430030, China; 3Department of Pharmacology, Shanghai Medical College, Fudan University, Shanghai 200032, China

**Keywords:** healthcare workers, perceived stress, family companionship, mental health, subjective happiness

## Abstract

**Aim**: This study aimed to understand the impact of perceived stress on the subjective happiness of Chinese healthcare workers (HCWs) and to further explore the chain-mediating role of family companionship and mental health. **Background**: In the face of tense doctor–patient relationships; a heavy workload; long working hours; seemingly endless shifts; potential professional title promotions; work performance assessments; and the difficult balance between family, work, and other aspects of life, HCWs are often under great pressure, which can endanger mental health and reduce subjective happiness. However, the role of healthcare workers’ active participation in family companionship in mental health and subjective happiness is not clear. **Method**: We used a mixed research design to collect data in two locations (Hospital A and Hospital B) in Wuhan, China. A self-distributed questionnaire was assigned to HCWs through the Research Electronic Data Capture survey. A total of 368 valid surveys were obtained. **Results**: Hospital A’s perceived stress level and mental health problems were more severe, while Hospital B had a higher subjective happiness score and more time to spend with their families. Subjective happiness was affected by children, education, occupation, health status, commuting time, and the scores of perceived stress and depression. The scores of perceived stress and mental health were significantly negatively associated with subjective happiness and family companionship, and there was a significant positive correlation between subjective happiness and family companionship. The results also showed that family companionship and mental health acted as serial mediators between perceived stress and subjective happiness. However, family companionship did not play a mediating role between perceived stress and subjective happiness. Most HCWs had work–family conflicts, and a high amount of work pressure and feelings of powerlessness and not having sufficient time were common when they accompanied their families. **Conclusions**: HCWs had a high level of perceived stress and psychological distress, and their subjective happiness score was lower than that of the general population. Many HCWs experienced negative emotions when taking care of their families. Only a small number of people had enough time to spend time with their families and perform more prominently in busier hospitals. More importantly, perceived stress can indirectly have an impact on subjective happiness through a chain-mediating effect of family companionship and mental health, and family companionship may not always promote subjective happiness unless mental health is maintained. Therefore, in the future, we can consider carrying out interventions based on family companionship and mental health among HCWs to promote the healthy and harmonious development of individuals, families, and hospitals.

## 1. Introduction

In the current industrial society, stress has become a crucial public issue, especially in hospital settings, where medical services are expected to be available 24 h a day. In fact, healthcare workers (HCWs) are considered one of the occupational populations that are most susceptible to work stress [1]. In confronting a heavy workload; long working hours; seemingly endless shifts; tense doctor–patient relationships; potential professional title promotions; work performance assessments; and the difficult balance between family, work, and other issues, HCWs are often under great pressure [2,3,4,5,6], which was especially the case during the COVID-19 pandemic [7].

Although stress is inevitable, the effects of stress on individuals may vary depending on different perceptions of stress. According to the transactional theoretical framework proposed by Lazarus and Folkman [8], perceived stress is caused by one’s perception of demand as threatening and acknowledging that personal resources are insufficient to meet that demand. While stress is a neutral word, persistent and excessive stress perception has been confirmed to be an important risk factor for both physical and psychological health. The prevalence of mental disorders among HCWs is higher than in other occupations [9]. As reported by a recent meta-analysis that involved 62,382 samples, the reported rate of depression, anxiety, and distress symptoms in HCWs was 21.8%, 26.9%, and 48.1%, respectively [10].

Previous studies revealed that perceived stress not only endangers mental health but also impairs our ability to generate positive emotions such as happiness [11,12]. Happiness is the subjective experience of joy, satisfaction, contentment, and fulfillment, which is most often considered to involve three elements: frequent positive effects, infrequent negative effects, and cognitive evaluations of life satisfaction [13,14]. In this materialistic society, happiness has become the ultimate goal of life and a major subject in the fields of psychology, economics, and policy [14]. Research suggests that perceived stress is the most important happiness killer, which even overshadows the influence of other major factors [15]. As a result, stressed HCWs are less likely to gain happiness than general populations [16].

Furthermore, studies revealed that perceived stress could hinder individuals from integrating into social networks, which further affects their mental health outcomes and subjective happiness [17,18,19]. The webs of a person’s relationships are referred to as social networks, and they serve a variety of functions, such as influencing others, providing company, providing support, and facilitating social comparison. Consistently, companionship with loved ones has been identified as an effective intervention in fighting mental disorders [20]. Hence, companionship and mental health may play a mediating role between perceived stress and subjective happiness. To the best of our knowledge, little research has explored the impact of HCWs’ active participation in family companionship. For now, with the deepening of medical and health reform and the further implementation of actions to improve medical services, many Chinese HCWs are confronted with having to manage work stress and unfilled family duties [21]. As described in a Chinese proverb, “Jia he wan shi xing”, which means that family harmony is the basis for success in any undertaking. Investigating the current status of family companionship and exploring its role in the mental health outcomes of HCWs may help us detect their specific needs and provide evidence for future workplace mental health interventions.

Therefore, this study carried out a mixed study in two general hospitals in China to describe the characteristics of family companionship among Chinese HCWs and, specifically, explore the mechanisms underpinning the relationship between perceived stress, family companionship, mental health, and subjective happiness.

Inspired by the above findings, we proposed the following research hypotheses:

**H1.** 
*Perceived stress is associated with mental health, family companionship, and subjective happiness;*


**H2.** 
*Family companionship mediates the relationship between perceived stress and mental health;*


**H3.** 
*Family companionship mediates the relationship between perceived stress and subjective happiness;*


**H4.** 
*Family companionship and mental health act as serial mediators between perceived stress and subjective happiness (as shown in Figure 1).*


In addition, we also described the current situation of HCWs’ perceived stress, family companionship, mental health, and subjective happiness in different general hospitals, explored the factors affecting HCWs’ subjective happiness, and finally carried out a qualitative analysis of open-ended survey responses to further understand the feelings of HCWs when caring for and spending time with their families.

## 2. Methods

### 2.1. Study Design, Settings, and Population

We used a concurrent triangulation design in the mixed research. A self-distributed questionnaire was assigned to HCWs through the Research Electronic Data Capture survey. At the end of the survey, there was an open-ended question (How did you feel about taking care of and spending time with your families?), and respondents were invited to answer it online. Responses varied from single-word answers to entire paragraphs. The purpose of open-ended survey responses was to explain and elaborate on the experience of HCWs in taking care of their families so as to further analyze the role of family companionship. The study was conducted from 11 August 2021 to 30 December 2021 in two Chinese hospitals, Hospital A and Hospital B, a general and a tertiary public university hospital located in Wuhan, Hubei Province, China. We targeted all HCWs, such as nurses, doctors, and other health professionals. The convenience sampling method was employed. The inclusion criteria were as follows: (1) were aged 25–55 years old (because most people in this age group have both work and family obligations, they especially face more perceived pressure and mental health conditions); (2) had normal reading and writing ability and understood the questionnaire content; (3) participated voluntarily; (4) had no serious medical conditions. The exclusion criteria were as follows: (1) had severe mental or physical disorders; (2) planned to leave the current position within 6 months.

### 2.2. Data Collection

Before the study began, the researchers explained the purpose and significance of the study to the participants. All participants obtained informed written consent. The participant would open the survey and read through the consent form. When they reached the end, they would have the opportunity to provide their information and sign their name if they agreed to participate. Subsequently, anonymous online questionnaires were distributed to all target populations by e-mail and took around 15 min to complete. In order to avoid “one person, multiple answers”, each device could only be used to answer once. Immediately after participants submitted their questionnaires, two trained clinical nurses checked the answers to each questionnaire to ensure that there were no missing data. Meanwhile, to prevent careless responding, we utilized various methods, including providing instructional messages before they responded to the survey, setting reverse scoring entries in the questionnaire, proctoring the process of responding, and rewarding a small gift after the survey completion. According to the previous literature reports [22], the standard deviation of subjective happiness of the general population in Hong Kong was 1.05, the test level α was taken as a bilateral, 0.05, and the allowable error was 0.16 (15%). By considering the 20% loss rate, the required sample size was 199. In total, 368 HCWs were included in the study, which meets the requirements. The formula for calculating sample size was as follows (α, type I error; σ, standard deviation; δ, allowable error):$$n=\frac{z_{\alpha }^{2}*\sigma^{2}}{\delta^{2}}$$

### 2.3. Ethical Considerations

The study was conducted in accordance with the Declaration of Helsinki and was approved by the Medical Ethics Committee of the Tongji Medical College, Huazhong University of Science & Technology, China (approval number: IRB # 2021S141).

### 2.4. Instruments

#### 2.4.1. General Information Questionnaire

The general information questionnaire was designed by the researchers. It included sociodemographic data (age, sex, body mass index, marriage status, the number of children, educational level, occupation, working experience, current shift work, weekly working hours, commuting time, family income, and liability status), health status, and lifestyle (exercise, self-reported sleep quality, smoking, drinking, diagnosed chronic disease, and self-perceived health status).

#### 2.4.2. Perceived Stress

Perceived Stress Scale-14 (PSS-14), developed by Cohen et al. [23], is used to measure the level of perceived stress, and the Chinese version was revised by Yang et al. [24]. The scale consists of 14 items with two aspects: feelings of tension and the sense of having a lack of control. Each item is scored on a 5-point Likert scale, from 0 (“Never”) to 4 (“Always”), and the range of the score is 0–56, with a higher total score indicating a higher level of stress. The composite reliability (CR) value of the scale in this study was 0.775. In previous studies reported by Yang et al., the questionnaire had good sensitivity and specificity when the cutoff point was set to 25.

#### 2.4.3. Family Companionship

Family companionship was assessed through a self-developed questionnaire with two questions: (1) Do you have enough time/energy to relax with your family every week? (2) Do you have enough time/energy to take care of your family every week? Each item is scored on a 3-point Likert scale, from 0 (“Not enough time/energy”) to 2 (“Have enough time/energy”). The final result is presented as an average, with a higher score indicating a higher level of family companionship. Construct validity of the questionnaire was tested using exploratory factor analysis (EFA). The items of family companionship were clustered into one factor, which could explain 77.7% of the total variance. The CR value of the scale in this study was 0.721, which meant that the reliability of this questionnaire was acceptable. The cutoff points were set to 0, 0.5–1.5, and ≥2 points.

#### 2.4.4. Mental Health

The Depression, Anxiety, and Stress Scale-21 (DASS-21), developed by Henry et al., is widely used to evaluate the mental health of individuals in the past week [25], and the Chinese version was revised by Jiang et al. [26]. This scale consists of 21 items with three aspects: depression (DASS-D), anxiety (DASS-A), and stress (DASS-S). Each item is scored on a 4-point Likert scale, from 0 (“Inconsistent”) to 3 (“Always consistent”). The sum score of each subscale is multiplied by two, and the range of scores is 0–42, with a higher score representing a lower mental health level. The CR value of the scale in this study was 0.902. The cutoff points of the depression subscale were set to 0–9, 10–13, 14–20, 21–27, and ≥28 points; the cutoff points of the anxiety subscale were set to 0–7, 8–9, 10–14, 15–19, and ≥20 points; the cutoff points of the stress subscale were set to 0–14, 15–18, 19–25, 26–33, and ≥34 points [27].

#### 2.4.5. Subjective Happiness

Subjective happiness was measured by the Subjective Happiness Scale (SHS), designed by Lyubomirsky et al. [28], and the Chinese version was validated by Nan et al. [22]. The scale consists of four items: three forward scores and one reverse score. Responses on a 7-point Likert scale ranging from 1 to 7 were summed and divided by 4, with higher scores indicating more happiness. The CR value of the scale in this study was 0.759.

### 2.5. Data Analysis

SPSS Statistics 24.0 (IBM Corporation, Armonk, NY, USA) and Amos 26.0 (IBM, USA) were used for data analysis, and the level of statistical significance was set at 0.05. Construct validity (EFA) and internal consistency (CR) were used to test the questionnaire. We used SPSS software to judge the distribution (skewness: |Sk| < 3; kurtosis: |Ku| < 10) and multicollinearity (variance inflation factor: VIF > 5) [29]. Data presented a normal distribution (skewness ≤ |0.96| and kurtosis ≤ |1.63|), and there was no multicollinearity (VIF < 0.95). Means and standard deviations of continuous variables, as well as the frequencies and percentages of categorical variables, are presented in descriptive statistics. A Student’s *t*-test, an analysis of variance, a chi-square test, and Fisher’s exact test were used to explore the differences between participants between groups. Multiple linear regression analyses were implemented to explore factors associated with SHS. The odds ratio (OR) of a 95% confidence interval was reported. The Pearson correlation coefficient method was used to understand the relationship between variables. The bootstrap method was used to test for the mediating effect. A total of 5000 preloads were made to calculate the chain-mediating effect of family companionship and mental health. The path analysis model based on the assumptions was built with the maximum likelihood estimation method. The absolute fit method used in this study featured the following parameters [30,31]: chi-square (*p* > 0.05), χ^2^/df (with a cutoff of ≤3), root mean square error approximation (RMSEA) (0.05 < cutoff < 0.08, good fitting), and standardized root mean square residual (SRMR) (with a cutoff of ≤ 0.08, acceptable). A cutoff value of ≥0.90 of the model fit indices, including the Goodness of Fit Index (GFI), Adjusted GFI (AGFI), Comparative Fit Index (CFI), Incremental Fit Index (IFI), and Normed Fit Index (NFI), was considered suitable.

After preliminarily reading the source materials, the single-word answers or paragraphs received from the open-ended questions were imported into the NVivo 12 software (QSR International Pty Ltd., Chadstone, Victoria, Australia), and the materials were coded, sorted, and analyzed. The software’s exploration function was used to query word frequency and automatically generated a word cloud map through a word cloud generator. It is a visual representation of unstructured text data, which is beneficial for displaying data and increasing comprehension [32]. Words were described by their size and color, and the size of the words was determined by their frequency of occurrence in the text, with larger words signifying a greater frequency. The minimum length of a word was three letters, and the grouping option was a complete match. The cloud did not include pronouns or conjunctions. Principles of confidentiality were strictly observed, and the responders were replaced by codes in the data analysis. A pragmatic approach to reflexive thematic analysis was used to analyze the open-ended survey responses [33]. Following a preliminary examination of the results, two researchers (DNF, a master in nursing, and QW, a doctor in nursing) independently coded the results. They took part in the study, were familiar with the research topics in the field of mental health and family companionship as they pertain to medical staff, and had prior experience working with medical personnel. We used the inductive process of iterative components in a complete qualitative study. After the initial coding, the data in each code were sorted and compared, and similar codes were classified according to the relevance of the topics, research objectives, and differences between concepts [34]. Finally, these categories were summarized into four topics, which presented the original data and conveyed significant features [35]. We also regularly discussed the data with members of the research team.

## 3. Results

### 3.1. Sample Characteristics

A total of 368 HCWs were included in this study, 217 in Hospital A and 151 in Hospital B. Most of the participants were married, female, and aged 30–39 years old. Among them, 76.4% were undergraduates, and 70.9% had one or more children. More than half of the participants were nurses (65.2%), had more than 10 years of work experience (50.8%), worked night shifts (79.3%), worked 35–44 h per week (54.9%), had a family monthly income of 10–20 thousand yuan (51.4%), and had liability status (80.2%). According to a recent China’s National Bureau of Statistics report, people with an annual income of USD 7.25–62.5 thousand (approximately 50–420 thousand yuan) are considered middle-class in China. In this survey, 51.4% of the participants’ family monthly income was at the level of 10–20 thousand yuan, which was in line with the middle-class group. The commuting time was 30–60 min. In terms of health status, about one-third were overweight, one-fifth suffered from chronic diseases, and nearly half exercised. Most people reported that their sleep quality was average or even poor, and most people felt that their health status was average. Group comparisons between Hospital A and B revealed no significant differences regarding sex, married status, educational level, working experience, commuting time, liability status, body mass index, diagnosed chronic disease, self-reported sleep quality, and self-perceived health status, but significant differences for age (*p* = 0.017), children (*p* = 0.021), occupation (*p* < 0.001), current shift work (*p* < 0.001), weekly working hours (*p* = 0.001), family monthly income (*p* = 0.007), and exercise habits (*p* = 0.047). Post hoc testing found that, compared with Hospital B, Hospital A had more participants that were aged <30 years old, had no children, and worked >44 h per week; however, Hospital B had more participants that were doctors, currently worked shifts, worked 35–44 h per week, had a family monthly income <10 thousand yuan, and exercised. Table 1 presents the demographic characteristics of Hospital A and Hospital B. Table 2 shows the categories of psychological health measurements.

### 3.2. Multiple Linear Regression

A multiple linear regression model was established to predict the related factors affecting the subjective happiness of HCWs. The final regression models explained 37.5% of the total variance in subjective happiness. The results showed that having children, having a postgraduate degree or above, feeling good in self-perceived health status, and having a 30–60 min commute (vs. >60) can positively predict subjective happiness, while for medical doctors and other HCWs, the PSS and DASS-D scores can negatively predict subjective happiness (see Table 3 for details).

### 3.3. Correlation Analysis

Pearson correlation analysis suggests that the PSS, DASS-D, DASS-A, and DASS-S scores were significantly negatively associated with subjective happiness and family companionship, and there was a significant positive correlation between subjective happiness and family companionship. This information is provided in Table 4.

### 3.4. Verification Analysis of the Chain-Mediation Effect of Family Companionship and Mental Health

The IBM AMOS 26.0 software was used to construct a structural equation model, in which the perceived stress, subjective happiness, family companionship, and mental health were considered independent, dependent, and intermediate variables, respectively. The maximum likelihood method was used for model fitting and correction to identify the significance of the direct, indirect, and total effects in a chain-mediation model. A bias-corrected percentile bootstrap (5000 replicate samples) was used to verify the chain-mediation effect model, and the results are shown in Figure 2.

After debugging, the significance probability value of the model was 0.056 (*p* > 0.05), indicating that the model can adapt to the sample data. Moreover, the model fit indices were good: χ^2^/df = 2.153, RMSEA = 0.056, 90% CI [0.000, 0.103], SRMR = 0.016, GFI = 0.990, AGFI = 0.960, CFI = 0.995, IFI = 0.995, and NFI = 0.990. The total effect of perceived stress on subjective happiness (c = −0.064, *p* < 0.05) was statistically significant, and the direct effect of the path was as follows: c’ = −0.047, *p* < 0.05. The mediating effect showed that the total indirect effect was composed of three paths, among which the mediating effect of mental health between perceived stress and subjective happiness was −0.017, and the chain-mediating effect of family companionship and mental health between perceived stress and subjective happiness was −0.001. The 95% CI corresponding to both paths did not consist of zero, indicating that the effect was significant, accounting for 26.6% and 1.6% of the total effect, respectively. However, the mediating effect of family companionship between perceived stress and subjective happiness was not significant (see Table 5 for details).

### 3.5. Results of Open-Ended Survey Responses

Of the 368 participants who were included in this study, 249 (67.7%) provided responses to the open-ended items and are included herein. Among the respondents, 86 (34.5%) had no time to spend with their families, 150 (60.2%) had little time, and 13 (5.2%) had enough time. The results of the word cloud are shown in Figure 3. The top five words that appeared most were “time”, “care”, “family”, “take”, and “children”. Four themes elucidating HCWs’ feelings, their current situation, and how to take care of their families in the future were identified in the data. This information is given in Table 6. Figure 4 shows the coding counts between different family companion groups, which indicated that most HCWs who had no time or little time to spend with their families reported having a lack of time, feeling powerless, and other negative emotions.

## 4. Discussion

Social support can significantly reduce the harmful effects of stress conditions and prevent individual mental problems [36]. Previous studies have shown that, among female medical staff, the indirect impact of work–family conflict on anxiety symptoms via emotional exhaustion can be moderated by social support [21]. Cross-sectional studies on the influencing factors of nurses and nursing students on subjective happiness have been conducted, and family support is one of these factors [37,38]. However, it is not clear how social support affects subjective happiness in HCWs at present. Therefore, in this mixed study, we included 368 HCWs from a Chinese general hospital and a tertiary public university hospital to explore the effect of participating in family companionship on subjective happiness as well as the relationship between perceived stress and mental health. The results showed that 73.4% had a high amount of perceived stress, slightly higher than that found by Chen and Yan et al. [39,40]. Only 5.2% of HCWs had enough time to spend with their families. In terms of mental health, 54.1%, 66%, and 49.5% of HCWs had different levels of depression, anxiety, and stress. These results are similar to those of Julien et al. [41] and are close to the second peak of COVID-19 [42], revealing that HCWs face severe psychological problems. In the 368 HCWs, the SHS score mean was 4.13 (SD 0.90), lower than that of the general population in Hong Kong [22]. The HCWs in this study showed differences in perceived stress, mental health, subjective happiness, and family companionship (*p* < 0.05). Post hoc testing also found that the perceived stress level and mental health problems of participants from Hospital A were more serious, while Hospital B participants showed a higher SHS score and had more time to spend with family, which may be related to their decreased shift work and the lower amount of hours worked per week.

Multiple linear regression showed that having children, having a postgraduate degree or above, feeling good in self-perceived health status, and having a commute time of 30–60 min (vs. >60) can positively predict subjective happiness, while for medical doctors and other HCWs, the scores of PSS and DASS-D can negatively predict subjective happiness. Most Asian countries believe that children will make people happier [43]. A study in Brazil showed that having children can significantly improve the happiness of psychiatrists [44]. Education can improve health status through a higher income, making better use of medical information, and understanding the potential harm caused by certain behaviors. It was also confirmed that higher education affects residents’ well-being and further improves their health status [45]. A good health condition, in turn, can also enhance subjective well-being [46]. There is evidence showing that people who spend a high amount of time commuting have lower subjective well-being, suggesting a “commuting paradox” [47]. Longer commuting times and greater commuting distances can have an impact on a person, not just in terms of physical health, but with regard to negative psychological and behavioral outcomes [48]. In a meta-analysis of the psychological effects of emerging virus outbreaks on HCWs, subgroup analyses revealed a higher prevalence of symptoms of anxiety and depression among nurses [49], who face greater work pressure and more severe psychological damage [50]. However, Negri et al. conducted a cross-sectional study on the job satisfaction and well-being of doctors and nurses involved in the management of multiple sclerosis in Italy and found that nurses scored higher on job well-being than doctors [51]. Our results also showed that nurses were happier than doctors and other professionals. In the future, more large-sample multicenter studies can be carried out to verify this. The negative effects of depression, anxiety, and stress on well-being were confirmed [52,53,54], and our research reached a consistent conclusion, albeit DASS-A and DASS-S were not significant.

Previous studies confirmed the positive effect of social support on mental health outcomes; however, little research has explored the impact of HCWs’ active participation in family companionship, which can be regarded as a kind of social support. Therefore, in this mixed method study, we tried to draw on the ideas of previous studies on social support based on our data. According to the result of Pearson correlation analysis, we found that perceived stress was positively correlated with mental distress and negatively correlated with family companionship and subjective happiness, which supports Hypothesis 1. By conducting the path analysis model, we found that family companionship acted as a mediator between perceived stress and mental health, which is consistent with Hypothesis 2. That is, the more stressful the HCWs were, the less family companionship they would enjoy, and thus the more mental distress they would suffer. This finding is similar to that of Bedaso et al.’s study, which demonstrated that a higher level of social support could buffer the negative impacts of perceived stress on mental health [18]. The results from our qualitative study provide an explanation for these findings. HCWs, due to long working hours, work overload, night shifts, and other phenomena, perceived a high level of work–family conflict, and they had less time and energy to spend with their families, resulting in a sense of guilt. In conclusion, work stress not only causes HCWs to perceive stress at work, but it also disrupts the balance between work and personal life, further hindering HCWs from sustaining their mental health.

It is worth noting that, although previous studies have shown that social support can play an important mediating role between stress and life satisfaction [55], we cannot draw that conclusion in this study. Furthermore, we did not find any significant mediating effect of family companionship on the association between perceived stress and subjective happiness, as described in Hypothesis 3. However, we verified that family companionship and mental health act as serial mediators between perceived stress and subjective happiness, which is consistent with Hypothesis 4. This finding sheds new light on the full mediating effect between family companionship and subjective happiness, implying that family companionship may not always promote subjective happiness unless mental health is maintained. A study conducted by Cichy et al. also concluded similar results [56], which revealed that extensive family networks could act as a double-edged sword: the family members who provide support may also have their own demands. The clinical implications for the above findings may be considerable. Interventions on encouraging active participation in family companionship, with a specific focus on reducing mental distress, may be beneficial in elevating the level of subjective happiness among HCWs.

## 5. Conclusions

In this study, we reported that HCWs had a high level of perceived stress and psychological distress, and their subjective happiness score was lower than that of the general population. Many HCWs had negative emotions in regard to taking care of their families. Only a small number of people had enough time to spend with their families and perform relatively prominently in busier hospitals. More importantly, perceived stress can indirectly have an impact on subjective happiness through a chain-mediating effect of family companionship and mental health; family companionship may not always promote subjective happiness unless mental health is maintained. Therefore, interventions based on family companionship and mental health should be carried out to promote the healthy and harmonious development of individuals, families, and hospitals. The organization should play a fundamental role in the integration between family life and work, e.g., by providing a supportive environment, allowing more personal time, and providing flexible and coordinated scheduling systems. Psychologically informed education and training can be carried out, and HCWs can be aided in establishing positive coping mechanisms in the face of pressure through face-to-face psychological counseling. Moreover, in terms of family, family members can be encouraged to show compassion and support and actively share in family responsibilities. HCWs should also master effective stress adjustment methods so as to help themselves cope with negative emotions, face work and family with greater enthusiasm, and further improve their subjective happiness.

## 6. Limitations

The limitations of this study are as follows: first of all, the results of the studies are based on a cross-sectional design and do not imply a causal effect of perceived stress on subjective happiness. Therefore, further testing in longitudinal studies is needed to determine the association between the strength of the relationships between different variables. Second, the sample size of this study was relatively small and could not fully represent all Chinese HCWs. We only collected data from Wuhan to test our hypotheses, which are geographically limited. The conclusion that family companionship and mental health play a chain-mediating role between perceived stress and subjective happiness needs to be further verified in other populations and countries. Moreover, the sample included in this study was primarily women. Considering that women play multiple roles in society, as mothers, children, workers, and so on, they are often prone to experience more negative emotions, such as stress and anxiety, and this may have affected the research results. Furthermore, this study was carried out during the COVID-19 pandemic. In the future, we can consider collecting questionnaires at different time points and comparing the results. Finally, since participants could voluntarily choose to participate in the open-ended survey, some of them did not express their feelings with regard to taking care of their families, which may mean that the theme range presented in this study is not comprehensive.

## Figures and Tables

**Figure 1 ijerph-19-12058-f001:**
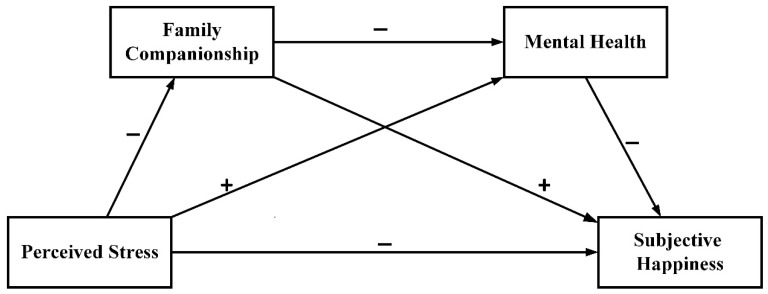
Hypothesized model of perceived stress, family companionship, mental health, and subjective happiness.

**Figure 2 ijerph-19-12058-f002:**
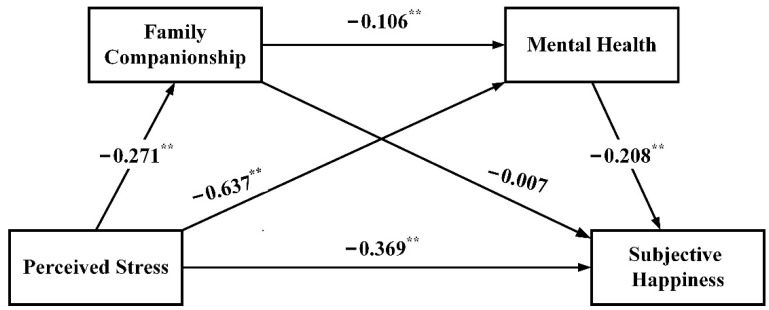
The role of family companionship and mental health as chain-mediators in the relationship between perceived stress and subjective happiness with standardized estimation (** *p* < 0.001).

**Figure 3 ijerph-19-12058-f003:**
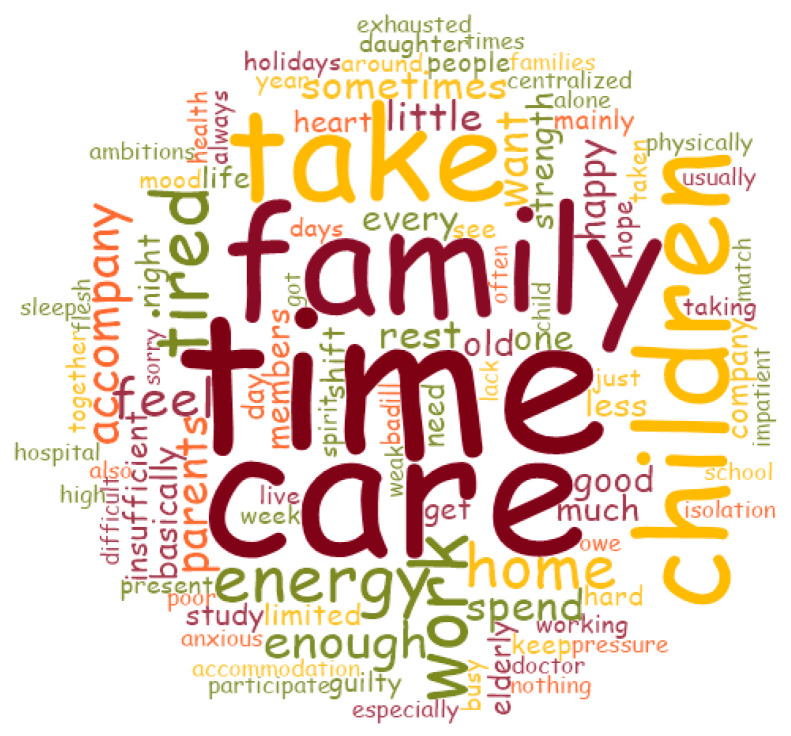
The word cloud of open-ended survey responses.

**Figure 4 ijerph-19-12058-f004:**
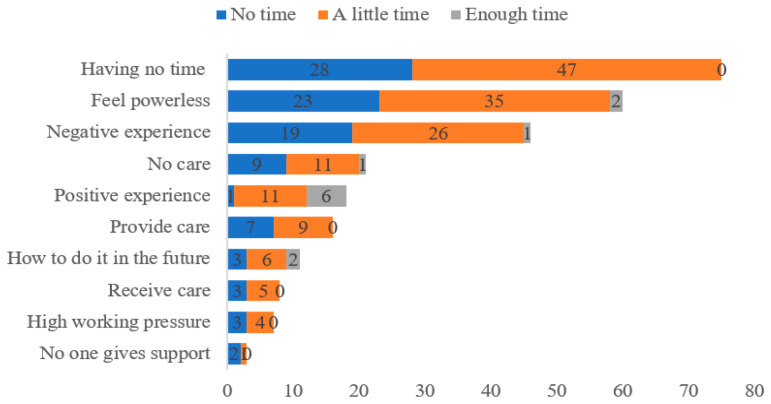
Coding counts between different family companion groups.

**Table 1 ijerph-19-12058-t001:** Demographic characteristics of Hospital A and Hospital B.

Characteristic	Category	Total (*n* = 368)	Hospital A (*n* = 217)	Hospital B (*n* = 151)	χ^2^	*p*-Value
Sex, *n* (%)	Male	55 (14.9)	30 (13.8)	25 (16.6)	0.523	0.470
	Female	313 (85.1)	187 (86.2)	126 (83.4)		
Age, year, *n* (%)	25–29	76 (20.7)	54 (24.9)	22 (14.6)	8.169	**0.017**
	30–39	223 (60.5)	130 (59.9)	93 (61.6)		
	40–55	69 (18.8)	33 (15.2)	36 (23.8)		
Marriage status, *n* (%)	Married	277 (75.3)	156 (71.9)	121(80.1)	3.250	0.071
	Unmarried	91 (24.7)	61 (28.1)	30 (19.9)		
Children, *n* (%)	One or more children	261 (70.9)	144 (66.4)	117 (77.5)	5.343	**0.021**
	No children	107 (29.1)	73 (33.6)	34 (22.5)		
Educational level, *n* (%)	Undergraduate degree or below	281 (76.4)	167 (77.0)	114 (75.5)	0.105	0.745
	Postgraduate degree or above	87 (23.6)	50 (23.0)	37 (24.5)		
Occupation, *n* (%)	Nurse	240 (65.2)	140 (64.5)	100 (66.2)	16.991	**<0.001**
	Medical doctor	86 (23.4)	41 (18.9)	45 (29.8)		
	Others	42 (11.4)	36 (16.6)	6 (4.0)		
Department, *n* (%)	Emergency/ICU/Operating room	78 (21.2)	49 (22.6)	29 (19.2)	0.860	0.651
	Outpatient/Ward	246 (66.8)	141 (65.0)	105 (69.5)		
	Auxiliary Department	44 (12.0)	27 (12.4)	17 (11.3)		
Working experience, year, *n* (%)	<10	181 (49.2)	106 (48.8)	75 (49.7)	0.024	0.877
	≥10	187 (50.8)	111 (51.2)	76 (50.3)		
Current shift work, *n* (%)	Yes	292 (79.3)	198 (91.2)	94 (62.3)	45.673	**<0.001**
Weekly working hours, *n* (%)	≤35	35 (9.5)	30 (13.8)	5 (3.3)	14.652	**0.001**
	35~44	201 (54.6)	105 (48.4)	96 (63.6)		
	>44	132 (35.9)	82 (37.8)	50 (33.1)		
Commuting time, min, *n* (%)	<30	102 (27.7)	63 (29.0)	39 (25.8)	1.889	0.389
	30–60	124 (33.7)	67 (30.9)	57 (37.7)		
	≥60	142 (38.6)	87 (40.1)	55 (36.4)		
Family monthly income, *n* (%)	<10 thousand yuan	56 (15.2)	22 (10.1)	34 (22.5)	12.118	**0.007**
	10–20 thousand yuan	189 (51.4)	114 (52.5)	75 (49.7)		
	20–30 thousand yuan	79 (21.5)	50 (23.0)	29 (19.2)		
	>30 thousand yuan	44 (12.0)	31 (14.3)	13 (8.6)		
Liabilities status, *n* (%)	Yes	295 (80.2)	178 (82.0)	117 (77.5)	1.156	0.282
Body mass index (kg/m^2^), *n* (%)	<23	243 (66.0)	148 (68.2)	95 (62.9)	1.110	0.292
	≥23	125 (34.0)	69 (31.8)	56 (37.1)		
Diagnosed chronic disease, *n* (%)	Yes	78 (21.2)	44 (20.3)	34 (22.5)	0.267	0.605
Exercise, *n* (%)	Yes	177 (48.1)	95 (43.8)	82 (54.3)	3.952	**0.047**
Self-reported sleep quality, *n* (%)	Poor	153 (41.6)	99 (45.6)	54 (35.8)	4.376	0.112
	Average	172 (46.7)	97 (44.7)	75 (49.7)		
	Good	43 (11.7)	21 (9.7)	22 (14.6)		
Self-perceived health status, *n* (%)	Poor	67 (18.2)	43 (19.8)	24 (15.9)	1.279	0.528
	Average	251 (68.2)	147 (67.7)	104 (68.9)		
	Good	50 (13.6)	27 (12.4)	23 (15.2)		

**Table 2 ijerph-19-12058-t002:** Differences in health measurements between the two locations.

Characteristic	Total (*n* = 368)	Hospital A (*n* = 217)	Hospital B (*n* = 151)	χ^2^/t	*p*-Value
**PSS-14, *n* (%)**					
Normal	98 (26.6)	42 (19.4)	56 (37.1)	14.328	**<0.001**
Health risks	270 (73.4)	175 (80.6)	95 (62.9)		
**Family companionship, *n* (%)**					
No time	114 (31.0)	75 (34.6)	39 (25.8)	8.200	**0.017**
Little time	235 (63.9)	136 (62.7)	99 (65.6)		
Enough time	19 (5.2)	6 (2.8)	13 (8.6)		
**DASS-21, *n* (%)**					
**Depression subscale**					
Normal	169 (45.9)	84 (38.7)	85 (56.3)	13.548	**0.009**
Mild	83 (22.6)	53 (24.4)	30 (19.9)		
Moderate	74 (20.1)	48 (22.1)	26 (17.2)		
Severe	34 (9.2)	25 (11.5)	9 (6.0)		
Extremely severe	8 (2.2)	7 (3.2)	1 (0.7)		
**Anxiety subscale**					
Normal	125 (34.0)	51 (23.5)	74 (49.0)	28.971	**<0.001**
Mild	41 (11.1)	25 (11.5)	16 (10.6)		
Moderate	113 (30.7)	74 (34.1)	39 (25.8)		
Severe	50 (13.6)	37 (17.1)	13 (8.6)		
Extremely severe	39 (10.6)	30 (13.8)	9 (6.0)		
**Stress subscale**					
Normal	186 (50.5)	81 (37.3)	105 (69.5)	40.849	**<0.001**
Mild	70 (19.0)	46 (21.2)	24 (15.9)		
Moderate	62 (16.8)	50 (23.0)	12 (7.9)		
Severe	38 (10.3)	30 (13.8)	8 (5.3)		
Extremely severe	12 (3.3)	10 (4.6)	2 (1.3)		
**SHS *, Mean (SD)**	4.13 ± 0.90	3.98 ± 0.88	4.35 ± 0.88	−4.033	**<0.001**

* A Student’s *t*-test was used.

**Table 3 ijerph-19-12058-t003:** The results of multiple linear regression for subjective happiness.

Variable	*β*	*t*	*p*	95% CI	Variance Inflation Factor (VIF)
Children (vs. no)					
Yes	0.297	3.569	**<0.001**	0.133~0.461	1.038
Educational level (vs. undergraduate degree or below)					
Postgraduate degree or above	0.244	2.217	**0.027**	0.027~0.460	1.585
Occupation (vs. nurse)					
Medical doctor	−0.345	−3.027	**0.003**	−0.569~−0.121	1.693
Others	−0.460	−3.672	**<0.001**	−0.707~−0.214	1.155
Self-perceived health status (vs. bad)					
Average	0.074	0.727	0.468	−0.126~0.274	1.632
Good	0.444	3.091	**0.002**	0.161~0.726	1.759
Commuting time (vs. >60)					
<30	−0.033	−0.349	0.727	−0.222~0.155	1.340
30–60	0.221	2.451	**0.015**	0.044~0.399	1.323
Family companionship (vs. no time)					
Little time	−0.067	−0.789	0.431	−0.233~0.100	1.201
Enough time	0.025	0.137	0.891	−0.337~0.388	1.211
PSS	−0.043	−6.023	**<0.001**	−0.057~−0.029	1.834
DASSD	−0.032	−3.939	**<0.001**	−0.048~−0.016	2.898
DASSA	0.012	1.237	0.217	−0.007~0.030	2.903
DASSS	−0.003	−0.405	0.686	−0.020~0.013	3.602

**Table 4 ijerph-19-12058-t004:** The correlation of variables used in the study.

	PSS	DASSD	DASSA	DASSS	SHS	Family Companionship
PSS	1					
DASSD	0.604 **	1				
DASSA	0.517 **	0.713 **	1			
DASSS	0.608 **	0.766 **	0.776 **	1		
SHS	−0.515 **	−0.478 **	−0.340 **	−0.421 **	1	
Family companionship	−0.271 **	−0.228 **	−0.233 **	−0.260 **	0.148 **	1

** *p* < 0.001.

**Table 5 ijerph-19-12058-t005:** Analysis of the chain-mediating effect of family companionship and mental health between perceived stress and subjective happiness.

	Path	Effect	Boot LLCI	Boot ULCI	*p*
Direct effect	Perceived stress → Subjective happiness	−0.047	−0.060	−0.032	**<0.001**
Indirect effects	Perceived stress → Family companionship → Subjective happiness	0.000	−0.003	0.004	0.836
	Perceived stress → Mental health → Subjective happiness	−0.017	−0.029	−0.007	**<0.001**
	Perceived stress → Family companionship → Mental health → Subjective happiness	−0.001	−0.002	0.000	**0.003**
Total effect	Perceived stress → Subjective happiness	−0.064	−0.075	−0.053	**<0.001**

**Table 6 ijerph-19-12058-t006:** List of themes and subthemes of the open-ended survey responses.

Theme	Codes	Sample Quotes
Work–family conflict	High working pressure	“There is a lot of work pressure and psychological pressure, so there is no way to get along with family under pressure”. (Participant 202)
		“Work intensity is high; aside from cooking, there is basically no energy to take care of children after work”. (Participant 347)
		“We have no holidays, and our working hours are so tight that it is difficult to adjust”. (Participant 148)
	Feel powerless	“I have two children. My eldest son is now 9 years old, and my youngest daughter is 8 months old. Taking care of these two children makes me physically overdrawn and mentally exhausted, and I lack sleep. I also easy become impatient when I take care of them after work”. (Participants 190)
		“Housework and children are taken care of by other family members. After finishing work, I feel I have no energy to take care of them”. (Participant 279)
		“There is little time for my children and family. Sometimes, even if I rest, I have to attend to meetings and my studies, or I feel so tired that I can’t take care of my family”. (Participants 345)
		“I’m too tired to go to work, and I can’t slow down after a night shift every three days, so there’s no way to take care of my family”. (Participant 271)
	Having no time	“My parents are old and in poor health. I’m busy at work. If my parents are ill, it’s difficult for me to spare enough time to accompany them when they need to see a doctor”. (Participant 58)
		“I want to, but I don’t have time. Every time I go to work, my children haven’t woken up yet. Most of the time they fall asleep when I come home from work”. (Participant 169)
	No one gives support	“There is no way to provide my children with the rich life and good family education environment I want, and no one can give me life and financial support, so I can only rely on myself”. (Participant 235)
Emotional experiences with respect to family members	Positive experience	“Home is the warmest place. Although I feel tired every time I go to work, I will be very happy when I get home, especially when I see my children”. (Participant 264)
		“High-quality parent–child interactions make me feel happy” (Participant 259)
		“Even if I feel tired sometimes, I am still very happy to spend time with my children”. (Participant 157)
	Negative experience	“I have to work 5–6 days a week, have a teleconference 1–2 times in the evening, and take part in studies inside and outside the hospital during my rest time. Basically, if my family gets sick, they go to see a doctor by themselves. I have no time to spend with them, so I feel guilty”. (Participant 119)
		“I am a very unqualified daughter, wife, and mother”. (Participant 319)
		“I feel worried and anxious because I can’t take good care of my family”. (Participant 338)
		“I’m very worried because my children are disobedient and don’t study hard”. (Participant 134)
		“When I take care of children after work, it is particularly easy for me to lose temper and become impatient”. (Participant 52)
		“At present, I have worked in a fever clinic for almost a year, and I have been in centralized isolation accommodation and home isolation for nearly half a year. During this period, I have had no way to take care of my family and children, no normal social intercourse, no way to go out, and no way to spend time with children doing outdoor activities or accompanying them on their commute to or from school, and I can’t get together with relatives and friends, which makes me feel very helpless”. (Participant 192)
Family care relationships	Provide care	“My family is not around, and I am usually busy with work, so there is very little company, care, and greetings for my family, and there is not enough support for each other”. (Participant 108)
		“Rest time with my family is inconsistent, so we can’t spend time with children together”. (Participant 339)
		“Usually I take more care of my children”. (Participant 65)
	Receive care	“I have always been taken care of by my family”. (Participant 361)
	No care	“My family and I don’t live together, so my family doesn’t need much care from me”. (Participant 329)
		“My family thinks that taking good care of themselves is the best care for them”. (Participant 362)
		“At present, I am single, and everything at home is taken care of by my elders”. (Participant 137)
How to take care of family in the future		“In the future, I will maintain a good mood, take care of my family more, and bring them more warmth”. (Participant 86)
		“The parents are old, and the children not only care about their studies but also their life. I hope I can spend more time with them in the future”. (Participant 173)
		“I will try to balance the relationship between work and family”. (Participant 181)

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
