# Peer review of "The Effect of Perceived Stress, Family Companionship, and Mental Health on the Subjective Happiness of Chinese Healthcare Workers: A Mixed Research Method"

_ijerph, 2022, doi:10.3390/ijerph191912058_

Round 1
Reviewer 1 Report
Thank you for giving me the opportunity to read this paper which touches upon important questions of mental health and family companionship as a resource for HCW. My impression is, that there are a lot of interesting aspects in your data and results, but the paper needs to be reworked to make this more visible.
Introduction
I recommend that you sharpen the concepts of stress that you are using in your introduction.
You define stress as something negative in your definition in the first sentence, but then seem to switch to a more ambivalent concept of stress as positive and negative stress.
Futhermore, you seem see stress as "determined by individual subjective perception". Established models of psychsocial stress such as the job demand - control - modell or the effort-reword-imbalance model show that individual conditions are one part of perceived stress, but further, more structural dimensions also play a major role. The criticism of concepts such as "resiliance" have also made clear that (working on) individual perception does not solve structural causes of negative stress and the negative stress resulting from these causes. I think, refining your concepts would help to sharpen your introduction.
Methods
I was suprised by inclusion criteria (1) age 25-55 years old and (4) no serious medical conditions. Could you please elaborate a bit more on your decision to give readers a better understanding of your reasoning behind the inclusion criteria?
Data collection: Quantitative methods are not my core field of expertise, but it seems a bit odd to me that you used different ranges in the Lickert scales, especially as you are working with scores?
The open questions are not mentioned in the data collection. Please add a few sentences: Where did research participants have the opportunity to answer open questions and what were the questions?
As it remains unclear how qualitative data were collected, the paragraph about the analysis of qualitative data comes a bit abrupt. I recommend considering the following questions when reworking the section: As online questionnaires were used, I expected the qualitative data to be written data - but why would you need to transcribe it then? What is the structure of the data - long paragraphs, few words, a mix? How are the identities of the researchers/coders in relation to the research participants (e.g. are the authors also HCW?)? Which kind of thematic analysis did you use (please provide a source)? How did you use the word clouds for data analysis? Why did you work inductively, but without an iterative process?
Results
As some of your readers are not familiar with Chinese incomes, please provide a short sentence for context in line 2022: Is the family income a decent income or rather low?
You mention group comparison in line 206, but it remains unclear, which groups you built. Line 2012 let's me assume your groups are hospital A and B? Why did you build hospital A and B as groups and how does this add to the analysis and your research questions? Besides that, your research questions and hypotheses do not call for this group comparision. I recomment rephrasing the research questions/hypotheses or leaving the comparison our. Please provide a bit more information.
As mentioned above, quantitative methods are not my core field of expertise, so I cannot evaluate the quality of the calculations and results. The result section reads a bit unstructured though.
The section about the qualitative results relies too much on the table 6 and cannot properly be understood without the table. Unfortunately, table 6 is not displayed to me, so I don't know, if quotes and deeper analysis are provided there. Quotes from the data are important to give readers a better understanding of the data and the quality of analysis. With the current quality of analysis, the reader is left with the impression that nothing surprising or new was found through the qualitative data. Please try to give a little more information and also heterogenity to your results section, to give the readers a better understanding of the experiences of HCW in your sample.
Discussion/Conclusion
In line 271, you write that this is an observational study. This does not fit the chosen study design. Please rephrase, e.g. in mixed methods study or exploratory study.
Your discussion touches on many relevant points in regards to the mental health of HCW. Nevertheless, the discussion will need a better structure, so that the results of your study and your analysis can be fully seen. My suggestion: You established hypotheses in the introduction, but did not refer to them in the results section and only mention them shortly in your discussion (line 317ff.). You could structure the discussion following your hyptheses and integrate your results and the existing literature.
Limitations
Point 3 in your limitations reads a bit confusing - would it be better, if others reported the experiences of HCW? Also, please elaborate a bit more on the potential bias (line 368) - do you mean a positive selection bias?
Language
I recommend to do a language check.
Author Response
Dr. Reviewer,
thank you very much for your suggestion. We sincerely appreciate your constructive and positive comments. I responded to your comments point by point and uploaded the file as attachment. If you have any suggestions or questions about this article in the future, please feel free to contact me. Thank you and best regards.
Yours sincerely,
Huang Sufang

Reviewer 2 Report
The paper presents an interesting, pertinent, and current issue. The work is targeted to a larger audience of the journal and gives considerable contribution to the advancement of scientific knowledge.
This paper has several important strengths, such as the relevance of the topic in the field of research; the rigour presented in the methods section following the steps of the research process; and the detailed presentation, namely the tools used, the robustness of the data and its connection to the conclusions and limitations presented.
In sum, this was a very interesting paper to read.
However, some aspects need further clarification, and the following points would benefit from additional information:
Abstract:
The conclusion must refer to the produced outcomes highlighting the study findings and not just suggestions/implications resulting from the Research.
Introduction:
Presents good articulation of content but should clarify in a more robust way concepts integrated in the research, such as happiness and family companionship. Some of the references used are not updated to the most recent scientific evidence and need revision (e.g. 1993, 2009).
Figure 1 should consider the relations addressed in the hypotheses, thus providing a more complete and accessible interpretation to the reader.
Data collection and Ethical Considerations:
‘Two trained clinical nurses supervised the completion of the questionnaires.’ This information is not clear. The participants answered in the workplace, were they from it (being online), does the supervising presence of the nurses not compromise the answers?
How long was the questionnaire available? What is the precise period of data collection?
Since a ‘Research Electronic Data Capture survey’ was applied, how was the informed consent of the participants obtained?
You mention ‘Principles of confidentiality were strictly abided, and the responders were replaced by codes in the data analysis.’ You should include here some examples of the codes attributed; it gets confusing when in Table 6 you mention e. g. "Participant Zhao" Zhao is the participant's name? How is confidentiality guaranteed?
Results:
Table 1 presents age classes (< 30; 30-39; ≥40) considering the inclusion criterion ‘age 25-55 years old’. Age classes presented in this Table should be according to this information. Also, the symbols in this variable: ≤Undergraduate ≥Postgraduate are difficult to understand.
Discussion:
The main topics are discussed but references need revision and update according to the study focus.
Limitations:
Being the samples predominantly women could this not be viewed as a limitation of the study when it aims to study ‘family companionship’ considering the roles still played by women in society ‘multiple roles-worker-mother...’?
The fact that this research was conducted during the COVID-19 was not itself a limitation as well? And have you considered that this study can be used as a comparison in future studies?
References: Some references are not updated considering the theme under study. There are also names of journals written in capital letters while others use lower case letters.
Figures and Tables: These should be revised as some are not easy to read and interpret (e.g., Table 2 and Figure 2)
Author Response
Dr. Reviewer,
Thank you very much for your letter and advice. We sincerely thank you for your constructive and positive comments. I replied to your comments one by one and uploaded the files as attachments. If you have any suggestions or questions about this article in the future, please feel free to contact me.
Thank you and best regards.
You sincerely,
Sufang Huang
